# Effects of segmentation errors on downstream-analysis in highly-multiplexed tissue imaging

**Matthias Bruhns** [ID][1,2,3,4,5], **Jan T. Schleicher** [ID][1,2,3,4], **Maximilian Wirth**[1,2,3,4], **Marcello Zago**[1,2,3,6], **Sepideh Babaei**[1,2,3,4], **Manfred Claassen** [ID][1,2,3,4]*

**1** Department of Internal Medicine I, University Hospital Tübingen, Tübingen, Germany, **2** M3 Research Center, University Hospital Tübingen, Tübingen, Germany, **3** Department of Computer Science, University of Tübingen, Tübingen, Germany, **4** Institute for Bioinformatics and Medical Informatics, University of Tübingen, Tübingen, Germany, **5** Quantitative Biology Center (QBiC), University of Tübingen, Tübingen, Germany, **6** Hertie Institute for Clinical Brain Research, University of Tübingen, Tübingen, Germany

* manfred.claassen@med.uni-tuebingen.de

**Data availability statement:** The code used to run experiments and generate figures, as well as the underlying ground truth data set, is available on https://github.com/mbruhns/SegmentationErrorBenchmark.

## Abstract

Highly multiplexed single-cell imaging technologies have revolutionized our ability to capture spatial protein expression at the single-cell level, thereby enabling a deeper understanding of tissue organization and function. However, these advancements rely on accurate cell segmentation, which defines cell boundaries to generate expression profiles. Despite its importance, there is a gap in quantifying how segmentation inaccuracies propagate through analytical pipelines, particularly affecting cell clustering and phenotyping. We introduce a framework that uses affine transformations to simulate realistic segmentation errors. Our approach mimics the variations induced by segmentation algorithms, allowing us to evaluate the robustness of downstream analyses under controlled perturbation conditions. We show that even moderate segmentation errors can significantly distort estimated protein profiles and disrupt cellular neighborhood relationships in feature space. Effects are most pronounced in clustering analyses, where both unsupervised k-Means and graph-based Leiden algorithms exhibit reduced consistency with increasing perturbation — especially with smaller neighborhood sizes. Similarly, cell phenotyping via Gaussian Mixture Models is adversely impacted, with higher levels of segmentation error leading to notable misclassifications between closely related cell types. These results highlight the importance of ensuring high-quality segmentation and careful data processing strategies to mitigate spurious results for downstream analysis tasks. Considering segmentation inaccuracies, possibly in a probabilistic modeling framework, will improve the reliability and reproducibility of findings in multiplexed tissue imaging studies.

**Funding:** M.B. was supported by DFG EXC 2064 and DFG EXC 2180. M.B. and M.W. were supported by HORIZON-MISS-2023-CANCER-01-01 101136622 (THRIVE). J.T.S. was supported by DFG CL 792/1-1. M.Z. was supported by the Else Kröner Fresenius Stiftung (ClinBrain). S.B. was supported by TUEAI-2023-ATF01. The funders had no role in study design, data collection and analysis, decision to publish, or preparation of the manuscript.

## Author summary

Finding the outlines of cells in an image, known as segmentation, is a crucial step for measuring protein levels in tissues. It is well known that current methods used for this task are inherently imperfect. In our work, we explore how segmentation errors affect the downstream analysis of tissue images. We present a benchmark that introduces synthetic defined and yet realistic segmentation errors. This approach allows us to examine how even modest errors alter the definition of single-cell expression profiles, cell groupings and phenotyping. For instance, standard methods for grouping cells become less reliable as errors increase, and in particular distinguishing between related, as well as distant cell types proves more challenging. Our study highlights the need for high-quality cell segmentation and the choice of suitable evaluation metrics. By understanding and accounting for segmentation errors, we expect multiplexed tissue images to better contribute to reliable and reproducible insights into how tissues confer function, as well as dysfunction.

## Introduction

Highly multiplexed single-cell imaging technologies have revolutionized spatial proteomics by allowing the simultaneous examination of multiple protein markers within tissue sections while preserving their spatial context. They enable a comprehensive understanding of cellular heterogeneity, tissue organization, and the role of the microenvironment in health and disease [1–3]. Among the leading techniques in this field are CO-Detection by Indexing (CODEX) [4], Multiplexed Ion Beam Imaging (MIBI) [5], MACSima [6], and Imaging Mass Cytometry (IMC) [7]. These methods employ different strategies for multiplexing and readout — ranging from repeated staining and imaging cycles (CODEX, MACSima) to heavy metal-conjugated antibodies analyzed by mass spectrometry (IMC, MIBI) — resulting in variations in resolution, number of markers, and data quality. Despite these technical differences, each platform produces high-dimensional images with one channel per labeled target, forming a rich dataset that requires extensive data processing to extract meaningful insights. In this work, we focus specifically on data generated by CODEX.

A typical data-analysis pipeline for multiplexed imaging includes preprocessing to correct for channel and batch effects, quality control to filter out low-quality images, segmentation to delineate individual cells, and downstream analyses such as clustering and neighborhood analysis. Segmentation is the most fundamental of these steps: Features for a single cell are generated by aggregating pixel-level intensities within the segmented region, linking protein expression data to specific cell phenotypes. Multiple segmentation algorithms have been proposed, including watershed-based approaches and more sophisticated deep learning methods [8]. Tools such as Mesmer [9], CellSeg [10], Cellpose [11], Stardist [12–14], UnMICST [15], MIMO-NET [16] and MICRO-NET [17] often perform well in benchmarks but remain prone to errors, especially in complex tissue environments. Past benchmarking studies have revealed significant variability in segmentation performance across different tissue types and imaging modalities [18,19]. Although these discrepancies are well-documented, the extent to which segmentation errors propagate through subsequent analyses — such as clustering and neighborhood analyses — has not been systematically evaluated.

To fill this gap, we aim to quantify the influence of segmentation errors on spatial single-cell analyses in multiplexed tissue imaging. Specifically, we simulate segmentation inaccuracies in multiplexed imaging datasets and assess how these distort downstream analyses,

including clustering and neighborhood mapping. By doing so, we provide a quantitative assessment for understanding the impact of segmentation quality on the overall interpretation of spatial single-cell omics data. This conceptual evaluation is expected to motivate others to consider our findings when performing similar analyses on multiplexed tissue imaging data.

## Materials and methods

### Perturbation of segmentation masks

Our perturbation procedure employs affine transformations to control the F1 score of a given segmentation mask. Specifically, we use the `Affine` function from the `Albumentations` library [20], which applies translation, rotation, scaling, and shearing to each cell mask. We specify interval input parameters for each sub-transformation and sample transformation parameters from corresponding uniform distributions, ensuring that each perturbed cell mask is similar but unique.

The procedure for processing an input mask involves the following steps: An image array is initialized to match the size of the input mask. Then, for each cell mask, the following steps are executed in sequence: (a) a sub-array containing that cell's mask is selected, (b) all pixels outside of the mask are set to zero (background), and (c) the sub-array is padded. (d) Combinations of affine transformations are then applied to the sub-array using the provided parameters. (e) Finally, nonzero pixels are written into their respective index at the output array. As these affine transformations result in fuzzy-shaped cell masks, we apply binary opening to each mask. A binary opening is defined as an erosion followed by a dilation (see S1 Text and S1 Fig for transformation definitions and examples).

To avoid the generated cell masks overlapping in some cases, we automatically detect touching cell masks and set these pixels to zero accordingly. This ensures a one-pixel separation border between all cell masks. For each perturbation strength considered, the parameter values were determined empirically (see S1 Text) by exhaustively searching in a pruned parameter space. During the perturbation process, individual cell masks can be overwritten by neighboring cells. To allow for cell-wise comparisons, we calculate the intersection of all cell IDs over all runs per perturbation strength. Additionally, we perform quality control regarding cell size and cell nucleus signal (HOECHST) to remove empty cell masks and incorrectly merged cells.

Algorithm 1 shows the entire procedure in pseudocode.

**Algorithm 1 Cell mask perturbation.**

```
 1: Create empty array output with the same size as input
 2: function PerturbSegmentationMask(input, parameters)
 3:    for each cell_id in the input do
 4:       Extract the sub mask S of cell with label cell_id
 5:       Set all pixels in S unequal to cell_id to zero
 6:       Apply zero-padding to S
 7:       Apply affine transformation to S with parameters
 8:       Save nonzero pixel values of S into output
 9:    end for
10:    for all masks that share a border do
11:       Remove border cells of one of the two cells selected at
   random
12:    end for
13:    return output
14: end function
```

## Quantification of segmentation quality

We utilize an Intersection-over-Union (IoU)-based F1 score, as used in [19]. The IoU (also known as the Jaccard index) is defined by Eq 1, where $N_g$ are neighborhoods of ground truth cells and $N_p$ their predicted counterpart. Following the standard established in prior publications, a cell is considered correctly segmented (true positive) if any other cell with an IoU greater than or equal to 0.5 with the reference cell exists. Cells with lower IoU matches are classified as false positives, while cells without matches are treated as false negatives. The F1 score is defined in Eq 2.

$$J(N_g, N_p) = \frac{|N_g \cap N_p|}{|N_g \cup N_p|} \tag{1}$$

$$F_1 = 2 \cdot \frac{\text{precision} \cdot \text{recall}}{\text{precision} + \text{recall}} = \frac{2 \cdot \text{tp}}{2 \cdot \text{tp} + \text{fp} + \text{fn}} \tag{2}$$

## Classifier two-sample test

We perform a classifier two-sample test to evaluate whether the perturbations introduce significant differences in the data distributions. This test involves training a classifier to distinguish between samples from two datasets. If the classifier cannot differentiate between them, the datasets are considered to come from the same distribution. We use the CatBoost classifier [21], a gradient-boosting algorithm. We classify original and disturbed datasets using five-fold cross-validation. High accuracy indicates that the datasets can be easily distinguished. An accuracy close to chance level suggests high similarity, implying that the perturbations did not introduce significant distributional changes.

## Neighborhood preservation analysis

We analyze the neighborhood change in feature (expression) space for each cell to investigate the change in features in a local context. To quantify this, we adapt the set difference view proposed by [22]. Instead of considering the difference, we calculate the intersection of neighborhood sets for each pair of data points, leading us to the set intersection view, shown in Eq 3. We compare the ground truth neighborhood-set $X$ for each cell $i$ in a $k$-nearest-neighbor (kNN) graph with its counterpart in the set $\tilde{X}$ of the perturbed data.

$$\text{JKNN}_k(i) = \frac{X_k(i) \cap \tilde{X}_k(i)}{X_k(i) \cup \tilde{X}_k(i)} \tag{3}$$

To minimize the introduction of additional uncertainties in the analysis, we rely on exact and not approximate neighborhoods by using a brute-force approach to calculate them. For efficient neighborhood calculation, we use the cuML library [23] together with Numpy [24] and just-in-time compilation via Numba [25]. Part of the preprocessing steps were done with CuPy [26]. We also include the kNN accuracy described by [27] for further analysis of neighborhood changes. This metric quantifies how often neighborhoods are from the same class.

## Clustering methods

We employ k-Means and Leiden clustering algorithms to evaluate the effect of segmentation errors on unsupervised clustering results.

**k-Means clustering.** k-Means clustering is a partitioning method that divides the dataset into $k$ clusters by minimizing the within-cluster sum of squares. Again, we use the implementation provided by the cuML library for efficient computation. We perform k-Means clustering on both the ground truth and perturbed datasets for varying values of $k$. Specifically, we consider values from 3 to 30, matching different levels of granularity for the expected number of cell types.

**Leiden clustering.** The Leiden algorithm [28] is a community detection method for graph clustering that improves upon the Louvain algorithm [29]. We construct k-nearest-neighbor (kNN) graphs from the data, where each cell is connected to its $k$ nearest neighbors in feature space. We vary the neighborhood size $k$ to assess the influence of segmentation errors on the clustering results. Here, we use the implementation provided by [30].

**Evaluation of clusterings.** The clustering results were compared using the Adjusted Rand Index (ARI) to quantify the similarity between the cluster assignments. The ARI is defined as:

$$\text{ARI} = \frac{\sum_{ij} \binom{n_{ij}}{2} - \left[\sum_i \binom{a_i}{2} \sum_j \binom{b_j}{2}\right]/\binom{n}{2}}{\frac{1}{2}\left[\sum_i \binom{a_i}{2} + \sum_j \binom{b_j}{2}\right] - \left[\sum_i \binom{a_i}{2} \sum_j \binom{b_j}{2}\right]/\binom{n}{2}} \tag{4}$$

where $n_{ij}$ is the number of objects in both cluster $i$ of the ground truth and cluster $j$ of the perturbed clustering, $a_i$ is the number of objects in cluster $i$ of the ground truth, $b_j$ is the number of objects in cluster $j$ of the perturbed clustering, and $n$ is the total number of objects.

## Gaussian Mixture Models for phenotyping

We assess the impact of segmentation errors on cell phenotyping using Gaussian Mixture Models (GMMs). A GMM models the data as a mixture of several Gaussian distributions with unknown parameters. We use the implementation provided by [31]. This model mimics the hierarchical gating approach that is used to assign cell types based on marker expression. The gating strategy is typically visualized as a tree structure recapitulating the cell type hierarchy (see S3 Fig).

We quantify the severity of incorrect phenotyping by modifying the [32] measure to a distance. With this we can quantify how distant two different phenotypes are considering our gating strategy. Given two nodes $X, Y$ on a tree $T$ we consider their distance as

$$d_{\text{WP}} = 1 - \frac{2N}{N_X + N_Y} \tag{5}$$

with $N_X, N_Y$ being the distance of both nodes to the root node and $N$ being the distance from the root node to the closest common ancestor of $X$ and $Y$, respectively.

**Balanced accuracy.** We utilize balanced accuracy to evaluate classification performance in phenotyping. It is defined as the average sensitivity (true positive rate) and specificity (true negative rate). This metric avoids inflated performance estimates that can occur with standard accuracy when one class strongly outweighs the others. The balanced accuracy is calculated as

$$\text{Balanced accuracy} = \frac{1}{2}\left(\frac{\text{tp}}{\text{tp} + \text{fn}} + \frac{\text{tn}}{\text{tn} + \text{fp}}\right) \tag{6}$$

## Results

We present a comprehensive analysis of the impact of segmentation errors on downstream analyses in multiplexed tissue imaging. We begin by detailing our in-silico perturbation study design and demonstrate the realism of the considered perturbations of cell segmentation masks. Following this, we assess the impact of segmentation inaccuracies on the structure of the result of various inference tasks, including changes in cell neighborhood in feature space, unsupervised cell type assignment, and phenotyping. S1 Fig summarizes the applied workflow.

### In silico segmentation perturbation study design

Our objective is to assess the propagation of segmentation errors on the outcome of downstream analyses. For this study, we evaluate an ensemble of segmentations of a dataset with varying magnitudes of segmentation error. This setting allows for the evaluation of the propagation of errors as a function of the magnitude of the segmentation error.

We implement a perturbation procedure designed to generate realistic perturbations for each cell mask within a given ground truth mask, as shown in Fig 1. The perturbation applied to each cell mask is based on a series of affine transformations, including scaling, translation, rotation, and shearing. Minimal examples are shown in S1 Fig. Following these transformations, a binary opening process is used to smooth the shapes of the masks. After each cell mask is perturbed, an additional procedure is used to ensure that the masks of any two cells do not touch. This is achieved by enforcing a minimum separation boundary of at least one

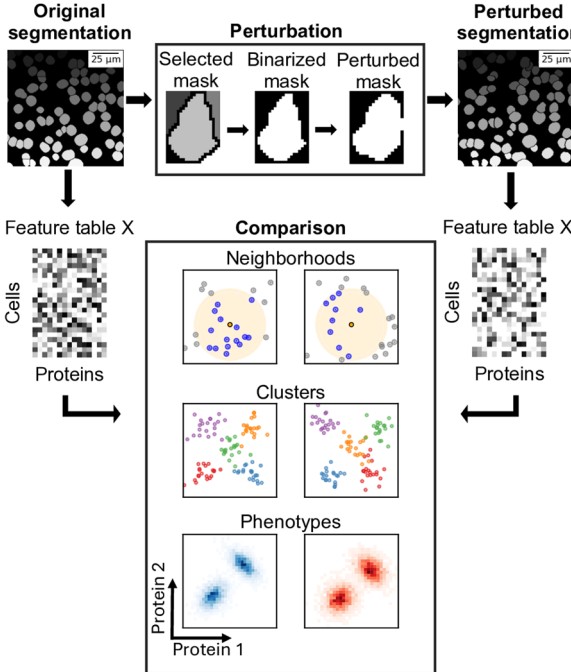

**Fig 1. Study overview**. Summary of the experiments. Starting from the original segmentation mask, the perturbation module generates a perturbed mask. For both masks, feature tables are generated. Based on these tables, further analysis and the corresponding results are compared.

pixel between adjacent cell masks. The parameters for each operation are sampled from specific uniform distributions that are empirically chosen for each perturbation strength. This empirical selection process also considers the average area change of the cell masks. We search a pruned parameter space to find suitable parameters and consider the resulting masks' area changes and F1 scores. This way, we ensure that the perturbations primarily affect the shapes of the cells rather than change their overall areas.

Calibrating the perturbation parameters ensures that the resulting segmentations retain high realism, allowing more accurate analysis of how segmentation errors affect downstream analyses. To evaluate the effect of segmentation errors on downstream results, we create modified masks at various perturbation levels. We define the perturbation level as the F1 score of the considered mask. Thus, lower perturbation levels correspond to more substantial perturbations and worse segmentations. Comprehensive benchmarking datasets with ground truth for full cell segmentations in multiplex tissue imaging are lacking due to the technique's novelty. However, datasets with manually segmented nuclei exist, such as the UnMICST dataset [15]. Here, we consider a manually curated segmentation mask generated by [33] as a proxy for ground truth segmentation to evaluate the F1 score-based perturbation strength of perturbed segmentation instances. The considered dataset consists of multiplexed tissue images of 15 hepatocellular carcinoma patient liver tissue samples. The HALO software, a toolbox including cell segmentation models, provided by Indica Labs, generated the corresponding masks (see also [33]). Here, we focus on the data of patient LHCC35 covering 234,958 cells for statistical analysis from three different tissue regions: healthy liver, tumor invasive margin, and tumor core. The marker panel of this study consists of 36 membrane markers and one nucleus marker. We consider perturbations as described above, yielding F1 scores of 70, 80, 90, and 100, respectively. Multiple perturbed segmentation instances are created for each perturbation strength to enable robust statistical analysis. On average, our perturbed datasets variants contain 201,168 cells (194725 cells (F1=70), 198953 cells (F1=80), 203384 cells (F1=90) and 210597 cells (F1=100), providing a basis for statistical assessment of the impact of segmentation errors on downstream analysis tasks (see Materials and methods for details).

## Affine transformations and binary opening generate realistic cell mask perturbations

We evaluate the realism of the perturbed segmentation masks. Specifically, we visually inspect and evaluate area distribution and expression profile bias. We first evaluate the realism of the perturbed masks through a qualitative visual inspection, as shown in Fig 2A. Specifically, we compare the visual appearance of the perturbed masks to the original ground truth segmentations to determine any noticeable artifacts introduced by the perturbations. Our observations indicate that the perturbed masks retain a visually consistent structure with the original masks, with no apparent irregularities or distortions like obviously unrealistic cell shapes with respect to curvature and overall shape. This suggests that the perturbation process did not compromise the visual integrity of the cell segmentation.

Next, we assess whether the perturbations distort the overall expression profiles of the cells by comparing the distributions of expression level differences across the entire cell population for both the perturbed and ground truth segmentations (Fig 2B). The comparison reveals that the expression profiles were only slightly altered by the perturbations, with most expression levels remaining consistent between the perturbed and original masks.

Additionally, we evaluate whether the perturbations introduce an artificial bias in cell size by comparing the cell area of the perturbed masks to that of the original segmentation. We

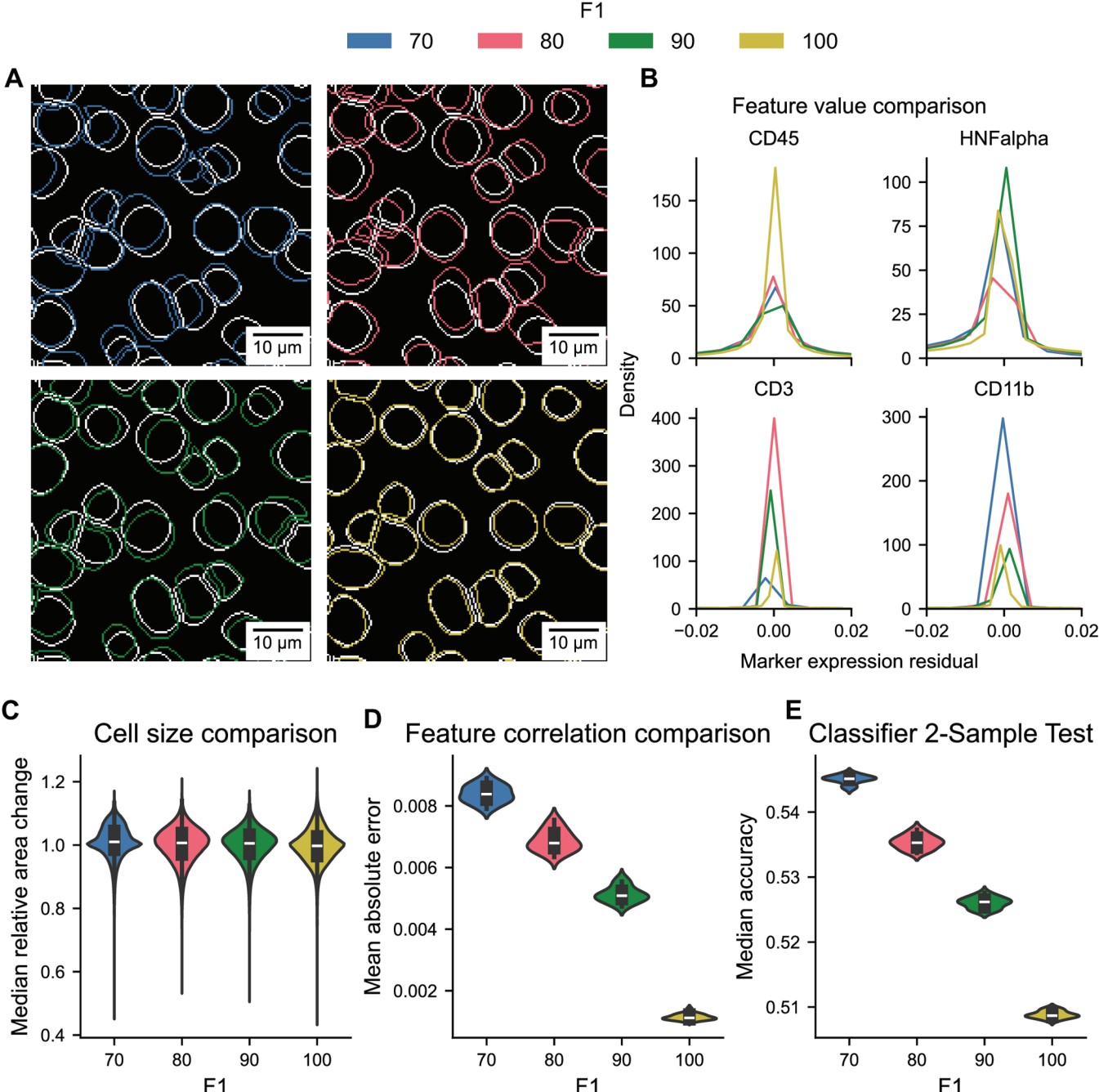

**Fig 2. Quality control of perturbations**. (A) Comparison of the original (white) and perturbed cell masks (colored according to perturbation strength: blue, red, green, yellow) based on mask borders on a randomly selected image slide. (B) Kernel-density estimation (KDE) for single-cell expression residuals between original and perturbed data. (C) KDE for median area changes of cells. (D) KDE for the mean absolute error between true and perturbed feature correlation matrices. (E) KDE for the results of the classifier 2-sample test.

compute kernel density estimates of the median ratio of cell areas in perturbed compared to ground truth segmentations to do this. The analysis shows that although the overall distributions are similar, there is a slight tail towards smaller cell areas (Fig 2C). The magnitude of

the effect is negligible, indicating that the perturbations do not introduce a significant bias regarding cell size.

To quantify the changes in correlation structure due to perturbation, we calculate the mean absolute error for the features' original and perturbed Pearson correlation matrices. These differences are minor, as presented in Fig 2D.

Finally, we conduct a classifier 2-sample test between the original and perturbed data to assess the segmentation perturbations' realism. The classifier 2-sample test evaluates whether two datasets originate from the same distribution by training a classifier to distinguish between them; higher accuracy indicates more distinguishable (and thus less similar) distributions. We train and test the classifier for all datasets and use all markers except the nucleus channel. This approach is motivated by downstream analysis, which typically does not consider the nucleus signal. The results in Fig 2E show that the classifier can distinguish between original and perturbed data only slightly better than random guessing, indicating the similarity of ground truth and perturbed segmentation masks.

In conclusion, the above evaluations show that the generated perturbed masks resemble the original masks regarding visual appearance, cell area distribution, individual expression correlation, and expression profile integrity. Therefore, these perturbations are realistic and suitable for benchmarking downstream analysis error propagation.

## Small segmentation errors distort the expression space neighborhood

We examine each cell's $k$-nearest neighborhood in the expression space to describe the change in protein expression. This analysis assesses how strongly perturbations affect the expression features concerning their effect on the k-nearest neighbor composition. Multiplexed tissue imaging data analysis is usually done on variance stabilized versions of the raw data. To account for this, we compare the change in the feature neighborhood. We apply the $\log(x + 1)$ transformation (also known as log1p) for variance stabilization. Additionally, we vary the number of neighbors used to construct the kNN-graph according to default values in single-cell analysis pipelines. The similarity of the neighborhoods is strongly affected by the strength of the perturbation (Fig 3A). The result shows that the neighborhood structure in the feature space is strongly affected even by segmentations that are close to the ground truth based on the IoU-F1 metric. We additionally aggregate over the perturbation strength (Fig 3B). It is apparent that the neighborhoods differ even for the weakest perturbations ($F1 =$

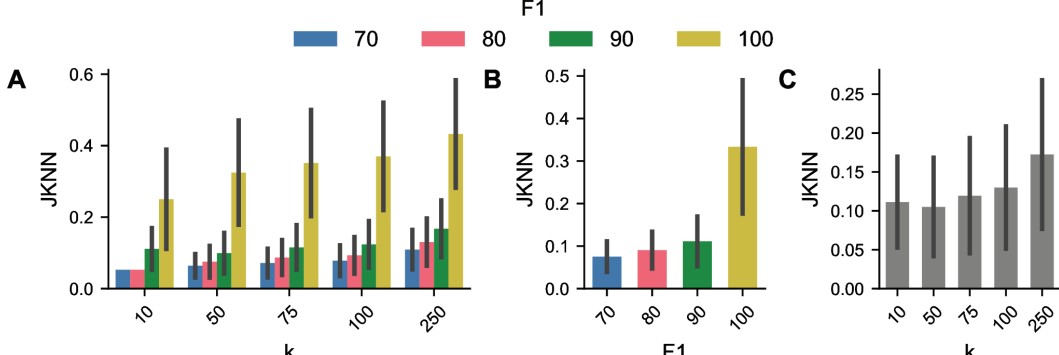

**Fig 3. Cell neighborhood comparison**. (A) Jaccard similarity of kNN sets (JKNN) for cells in different perturbation levels. (B) Aggregation of results for different perturbation strengths. (C) Aggregation of results for different numbers of neighbors. Error bars indicate one median absolute deviation.

100). An error-free segmentation as per the chosen metric still produces neighborhoods with less than 50% overlap. This result becomes more obvious when considering the distribution of intersection-over-union values for all cell pairs (S2 Fig). Since the overlap only needs to exceed 0.5, basically all cells considered as true positives might still only partially overlap with the ground truth segmentations. These results indicate the impact on further graph-based analysis, i.e., spectral clustering, as they explain how strongly generated kNN-graphs change under segmentation errors. Unsurprisingly, aggregation over neighborhood sizes (Fig 3C) shows an increase in similarity for larger neighborhoods.

## Perturbations have a progressive effect on unsupervised cell clustering

Cell type assignment is a fundamental analysis step for single-cell analysis. Typically, this involves using clustering algorithms that group cells based on similarities in their feature expressions. After clustering, each group is assigned a cell type identity based on the average expression of the features within that cluster. We consider k-Means and Leiden clustering as standard methods for clustering of this data type. To establish a baseline for our analysis, we use the ground truth feature tables to create cluster annotations for varying values of $k$. We then cluster all generated samples at each level of perturbation strength and compare them to the ground truth clustering using the Adjusted Rand Index (ARI) to quantify the similarity between two data clusterings. To account for randomness in the initialization step of both methods, we run each clustering five times with different seeds. We then consider the most similar cluster assignments for ground truth and perturbed data. This ensures that the observed differences are most probably due only to the introduced perturbations. Furthermore, we modify the preprocessing pipeline as in previous experiments, by considering both variance-stabilized and non-stabilized versions of the data (see Materials and methods for details). This consideration allows us to assess how segmentation inaccuracies in conjunction with preprocessing variants affect the downstream analysis of single-cell data. ARI decreases as perturbation strength increases (Fig 4A, 4C), indicating a progressive loss of correspondence between the clustering structures of original and perturbed mask features. ARI values vary across cluster numbers, but the general trend shows that clustering stability is higher for lower cluster counts. As the number of clusters increases, clustering consistency between original and perturbed masks decreases, as shown in (4D. Different preprocessing strategies modulate ARI values, with raw data maintaining greater consistency across perturbation levels (4B). The values range from 0.9 for the smallest number of clusters and weak perturbations to 0.5 for the opposite case. Overall, the results suggest that segmentation mask perturbations introduce non-negligible variation in clustering outcomes, with the effects becoming more noticeable at higher perturbation levels and cluster numbers. Additionally, these results show that even for F1 scores of 100, the result is cluster disagreement, highlighting the drawback of the default threshold of IoU-F1 as discussed in the previous section.

As an alternative clustering method, we investigate how segmentation errors affect unsupervised cell type assignment using the Leiden clustering algorithm, a popular method in single-cell omics commonly used for clustering analysis. Leiden clustering operates on a kNN-graph constructed from the data. We vary the neighborhood size $k$ to assess its influence on clustering robustness under segmentation perturbations, ranging from very local to more global connectivity graphs. Here we consider typical neighborhood sizes (10-100) for the kNN-graph for multiplexed tissue imaging studies, resulting in clusterings with cluster numbers ranging from 18-31 (Fig 5E). We apply Leiden clustering on raw and $\log(x+1)$-transformed data at each perturbation strength and compare the clustering results from the perturbed datasets to the ground truth using the ARI (Fig 5A, 5C). The adjusted rand index

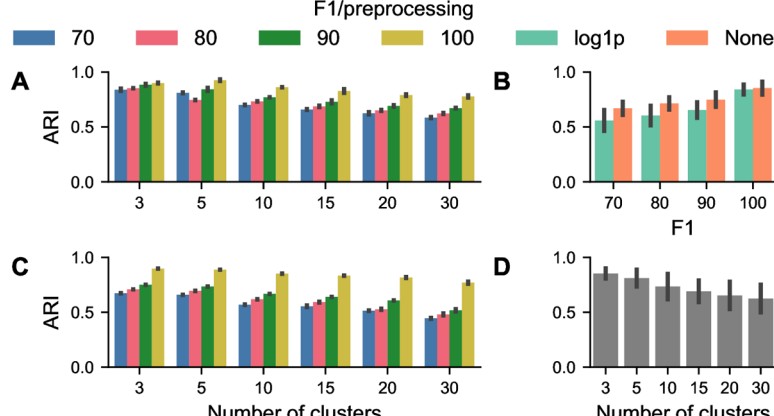

**Fig 4. Effect of segmentation error on k-Means clustering.** (A) Adjusted Rand Index (ARI) scores for different numbers of clusters. (B) Aggregations of ARI scores for perturbation levels. (C) ARI scores for different numbers of clusters for variance stabilized data. (B) Aggregations of ARI scores for perturbation levels for variance stabilized data. (D) ARI scores are aggregated for the number of clusters. Error bars indicate one median absolute deviation. Colors encode either perturbation strength or the applied preprocessing.

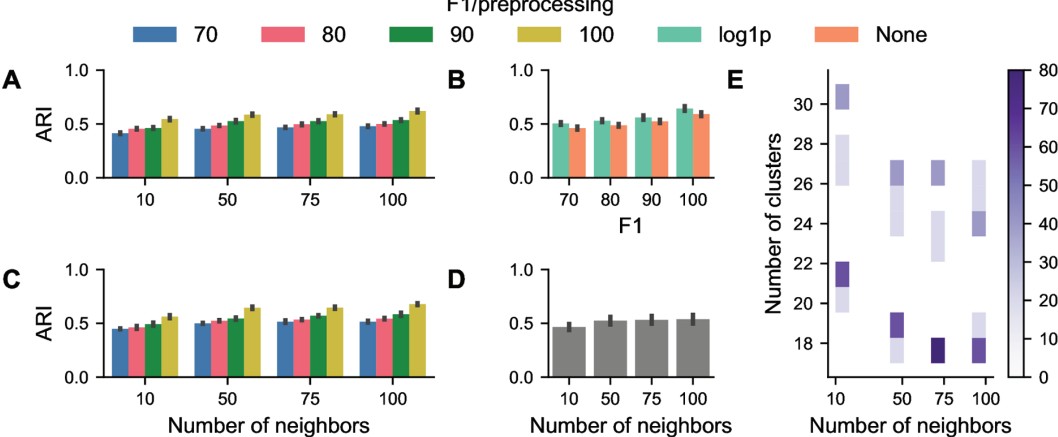

**Fig 5. Effect of segmentation error on Leiden clustering.** (A) Adjusted Rand Index (ARI) scores for different sizes of neighborhoods in the kNN-graph and perturbation strengths. (B) Aggregations of ARI scores for perturbation levels. (C) ARI scores for different sizes of neighborhoods in the kNN-graph and perturbation strengths for variance stabilized data. (D) ARI scores are aggregated for the number of neighbors. (E) Histogram showing the relationship between neighborhood size and resulting number of clusters. Error bars indicate one median absolute deviation. Colors encode either perturbation strength or the applied preprocessing.

decreases as perturbation strength increases, showing Leiden clustering is sensitive to segmentation mask perturbations (Fig 5B). The decline is gradual at lower perturbation levels but becomes more pronounced at more vigorous transformations. The number of neighbors has a negligible effect on clustering agreement, at least for values that fall in the typical range for typical single-cell clustering analyses from multiplexed tissue imaging data (Fig 5D). This result matches the observations from Fig 5E as the number of resulting clusters stays similar for all considered neighborhood sizes. Variance stabilization with $\log(x + 1)$ affects clustering robustness by preserving slightly higher ARI values. Specifically, the ARI values range from 0.65 for the largest neighborhoods and weak perturbations to 0.43 for the opposite case.

k-Means and Leiden clustering results show that ARI does not reach perfect agreement even when the IoU-based F1 score reaches 100. This discrepancy arises because the 50% IoU threshold allows poorly segmented cells to be classified as correct, meaning that substantial deviations in segmentation still influence the extracted features used for clustering. As a result, clustering assignments change despite the segmentation being considered "perfect" under the F1 metric. This suggests that ARI captures differences in features that the segmentation evaluation does not penalize, highlighting the drawback of low thresholds for segmentation evaluation. We also investigate segmentation aware probabilistic phenotyping with STARLING [34] and found similar sensitivity of clustering results to segmentation errors (S6 Fig).

## Segmentation errors cause inaccurate phenotyping

Here, our objective is to evaluate the impact of segmentation errors on the assignment of specific cell types. To this end, we use a three-component Gaussian Mixture Model (GMM) to assess positivity concerning cell type-specific protein marker combinations and to use these to replicate the gating strategy shown in S3 Fig to assign phenotypes to cells. Cells assigned to the first two components are considered to have low expression, while the third component encodes high protein levels (positive gating). Using three components reduces the number of false-positive gated cells and only reliably selects high-expression cells. Once the positive/negative status for each relevant marker is determined using these GMMs, the overall gating strategy combines these individual marker statuses to classify cells into specific types. For example, a CD8 T cell would be defined by a combination such as CD45 positive, CD3 positive, and CD8 positive. Across all perturbation levels, the balanced accuracy scores remain below 0.8, with lower perturbations showing slightly better concordance (Fig 6A). This suggests significant phenotype shifts, as approximately 20% of cells are assigned a different phenotype. We want to highlight that these incorrect assignments can be split into three categories: changes of subtype, granularity, and cell lineage. These error types differ in severity, with cell lineage changes being the most dramatic effects. Examples of all error types can be seen in Fig 6B, which shows a confusion matrix for F1=90. For example, 2% of CD4 T cells are incorrectly assigned to Tregs (subtype change), 4% of Immune cells get assigned to Parenchymal cells (cell lineage change) and 9% of LSECs are phenotyped as Parenchymal cells (change in granularity). One reason for the high number of wrong assignments for immune cells can be their smaller size compared to other cell types. Even small segmentation errors have a greater proportional impact when a cell has a smaller area.

To further quantify these different errors, we use the Wu and Palmer distance (see Eq 5)to assess changes in phenotypes for the shown confusion matrix ( Fig 6C). Zero values indicate no change, while a value of one indicates an assignment to an erroneous lineage. This analysis reveals that while the vast majority of cells get phenotyped correctly, 12% of cells suffer from errors of any kind, with nearly 5% changing their lineage.

## Change of phenotype labels in neighborhood

In addition to the analysis conducted in Small segmentation errors distort the 357 expression space neighborhood we use the generated phenotypes to get a more distinct reflection of change regarding neighborhoods in feature space. To do so, we calculate the kNN-accuracy. Fig 7 shows the results for this experiment. Unsurprisingly, the differences are less drastic than those in the JKNN comparison. While JKNN captures neighborhood changes per cell index, kNN-accuracy is less sensitive since it considers the most common phenotype in the neighborhood of a cell. Still, the results show changes in neighborhoods that again increase

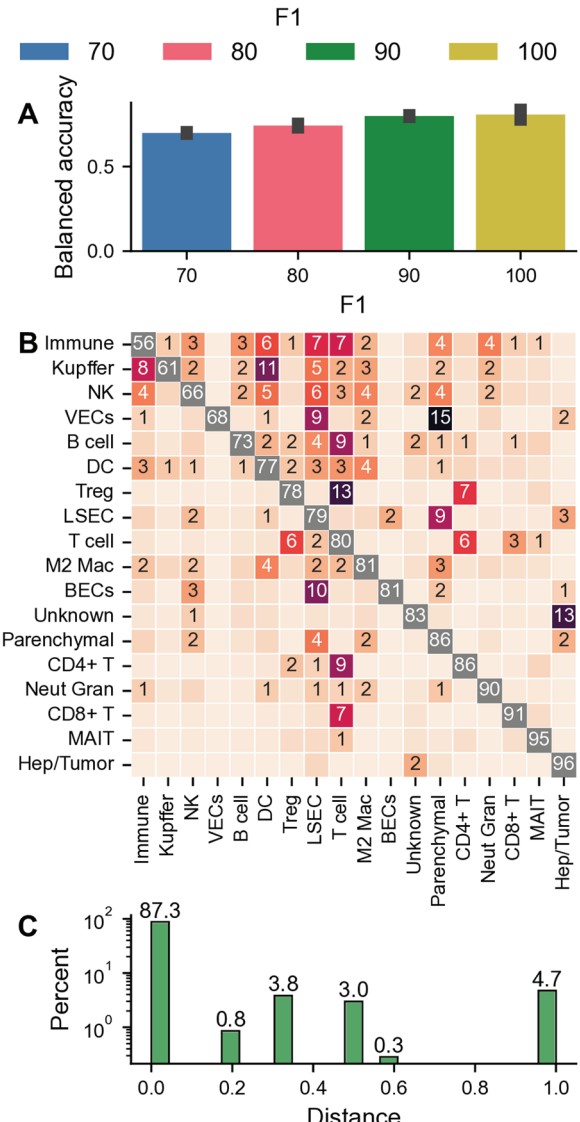

**Fig 6. Effect of segmentation errors on phenotype assignments.** (A) Balanced accuracy between ground truth and perturbed data. (B) Confusion matrix showing the errors for the Gaussian mixture model (GMM) based phenotyping. The shown perturbation is according to state of the art (F1=90). Only values ≥ 1 are shown. Color encodes the percentage of classified cells. For visual appearance, only off-diagonal elements are colored in. (C) Wu and Palmer distance for F1=90 based on the used gating strategy. Error bars indicate one median absolute deviation.

with perturbation strength and decrease with neighborhood size. These results match the findings from the JKNN analysis and the phenotyping results.

## Discussion

This study systematically examines how segmentation errors affect downstream analyses in multiplexed tissue imaging. We simulate segmentation inaccuracies by introducing realistic perturbations to cell segmentation masks and assess their impact on subsequent analytical

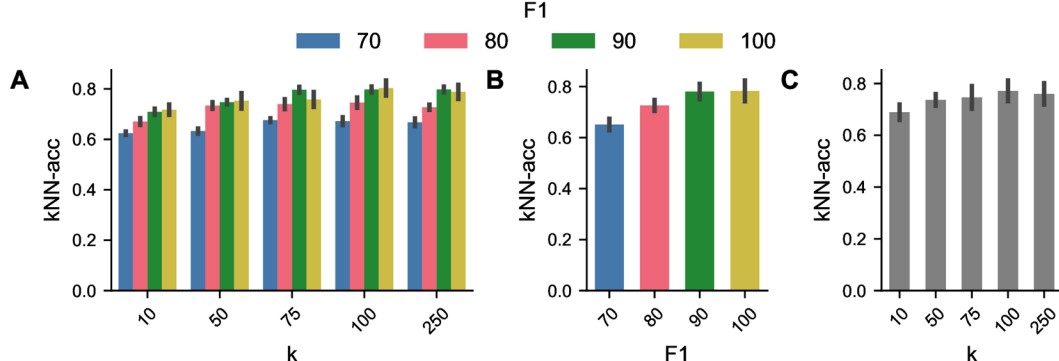

**Fig 7. Results of GMM-based phenotyping** (A) Median balanced accuracy when comparing phenotypes of ground truth and perturbed cells. (B) kNN accuracies aggregated over perturbation strengths. (C) kNN accuracies aggregated over neighborhood sizes.

tasks. The perturbations applied are based on specific affine transformations and may not capture all segmentation errors encountered in practice.

Our findings highlight the significant influence of segmentation errors on downstream results. Even minor perturbations lead to noticeable changes in estimated protein expressions, affecting the construction of cell neighborhoods in feature space. This was evident from the decreased similarity of kNN-graphs between the original and perturbed datasets as perturbation strength increased for both index and label comparison. Clustering and phenotyping were also sensitive to segmentation errors. The k-Means and Leiden clustering algorithms show reduced agreement with the ground truth clusters at higher perturbation levels. The Leiden algorithm, widely used in single-cell analyses, is significantly affected when smaller neighborhood sizes are used in the kNN-graph. In phenotyping tasks, segmentation errors decrease balanced accuracy scores and lead to notable misclassifications between specific cell types. We also show that errors in phenotyping not only include changes in cell subtypes but also range to changes in the inferred cell lineage.

The implications of these results are two-fold: First, the current usage of the IoU-F1 metric with a threshold of 50% to assess and benchmark segmentations should be interpreted with care. While it is tempting to use metrics that result in higher values, it is important to consider the implications of chosen thresholds. Second, segmentation errors can propagate through the analysis pipeline, leading to altered interpretations of cellular neighborhoods, misassigned cell types, and potentially misleading biological conclusions. A relevant real-world illustration can be seen with biomarkers such as the one developed by Salié et al. [35] for predicting response to immune checkpoint inhibitor (ICI) therapy. This biomarker relies on accurate counts of CD8+ T cells, B cells, and CD4+ T cells. An incorrect measurement of these cell populations could directly result in patients who would be excellent candidates for ICI therapy being overlooked and thus denied a potentially effective treatment. Researchers should be aware of the impact of segmentation inaccuracies highlighted in this study and consider strategies to assess and mitigate these effects. This issue has implications for the benchmarking of segmentation approaches. We suggest using stricter evaluation metrics to evaluate the performance of segmentation models, especially regarding the choice of underlying thresholds for the definition of true positive cell segmentations.

Specific multiplexed tissue imaging studies have to commit to a cell segmentation protocol. We have seen that there is a non-negligible impact of seemingly unavoidable segmentation errors on downstream analyses. This issue motivates conceiving future analysis pipelines that explicitly model the segmentation error introduced by the used models and propagate it through the full analysis pipeline. Such pipelines could translate similar work on related segmentation tasks of medical imaging data [36–38]. The work by Baumgarter et al. addresses this issue of segmentation errors with a hierarchical probabilistic model, which provides not one segmentation but a distribution of plausible segmentations. This way, any biological statement derived from segmentation-based single-cell features can be evaluated in a statistically sound way.

This study is based on a single dataset and several perturbations thereof. It serves as a proof-of-concept to showcase the underlying issue. Future research will expand on performing this perturbation analysis for further datasets. While we do not provide an installable software tool, our conceptual contributions and published code can act as guidelines for such analyses. As a broader framework, Nextflow [39] is a suitable approach for these kinds of studies. By designing the full analysis as a Netflow pipeline, each part of it can be implemented as a module which allows for specific configuration. A similar approach has been applied by Zappia et al. [40] to benchmark and compare feature selection for scRNA-sEq data.

## Conclusion

We demonstrate that segmentation errors in multiplexed tissue imaging can substantially impact downstream analyses, including cell neighborhood characterization, unsupervised clustering, and phenotyping. Given these insights, researchers should be aware of the potential propagation of segmentation errors through analytical pipelines, which can lead to misleading biological interpretations. Ensuring high-quality segmentation is essential for reliable results. Future work should focus on developing improved segmentation algorithms and error-aware methods to enhance the interpretability of biological findings.

In conclusion, accurate cell segmentation is vital for the reliability and reproducibility of findings in multiplexed tissue imaging studies. By recognizing and addressing the impact of segmentation errors, researchers can improve the robustness of single-cell analyses and advance our understanding of cellular behaviors in heterogeneous tissues.

## Supporting information

**S1 Text. Supplementary notes.**
(PDF)

**S1 Fig. Affine transformations**. (A) Minimal example showcasing the effect of single affine transformations on a simple shape. (B) The same transformations applied to a single cell segmentation mask.
(TIF)

**S2 Fig. Intersection over Union (IoU) of cells for considered F1 scores.** (A) Histogram over IoU values of cells. (B) Cumulative histogram. Any value above 0.5 will be considered a true positive (TP).
(TIF)

**S3 Fig. Gating strategy applied to the data**. Myeloids and lymphocytes can not be assigned but are shown for visual reasons.
(TIF)

**S4 Fig Confusion matrices for phenotyping for all perturbation strengths**. Only values $\geq 1$ are shown. Color encodes the percentage of classified cells. For visual appearance, only off-diagonal elements are colored in.
(TIF)

**S5 Fig Marker expression and segmentation masks on example tiles**. Each row shows a different tile, colors encode different marker expressions. The color of the mask outlines corresponds to perturbation strength. Empty masks and merged masks are filtered in later QC steps.
(TIF)

**S6 Fig Performance of STARLING clustering**. Each subplot shows the adjusted rand index for the initial k-Means clustering and after correction with STARLING. Error bars indicate one median absolute deviation.
(TIF)

## Author contributions

**Conceptualization:** Matthias Bruhns, Jan T. Schleicher, Maximilian Wirth, Marcello Zago, Sepideh Babaei, Manfred Claassen.

**Data curation:** Matthias Bruhns.

**Formal analysis:** Matthias Bruhns.

**Funding acquisition:** Manfred Claassen.

**Investigation:** Matthias Bruhns, Manfred Claassen.

**Resources:** Manfred Claassen.

**Software:** Matthias Bruhns.

**Supervision:** Manfred Claassen.

**Visualization:** Matthias Bruhns, Jan T. Schleicher.

**Writing – original draft:** Matthias Bruhns, Sepideh Babaei, Manfred Claassen.

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
