## [Decision Letter · Decision Letter 0]

1 Apr 2025

PCOMPBIOL-D-25-00324

Effects of segmentation errors on downstream-analysis in highly-multiplexed tissue imaging

PLOS Computational Biology

Dear Dr. Claassen,

Thank you for submitting your manuscript to PLOS Computational Biology. After careful consideration, we feel that it has merit but does not fully meet PLOS Computational Biology's publication criteria as it currently stands. Therefore, we invite you to submit a revised version of the manuscript that addresses the points raised during the review process.

Please submit your revised manuscript within 60 days Jun 01 2025 11:59PM. If you will need more time than this to complete your revisions, please reply to this message or contact the journal office at ploscompbiol@plos.org. Please include the following items when submitting your revised manuscript:

We look forward to receiving your revised manuscript.

Kind regards,

Guillermo Lorenzo

Academic Editor

PLOS Computational Biology

Pedro Mendes

Section Editor

PLOS Computational Biology

**Additional Editor Comments:**

Dear authors,

I would like to congratulate you on your work on the downstream impact of cell segmentation errors in highly multiplexed data presented in your manuscript, whose value has also been noted by the reviewers. Nevertheless, they have also identified several important issues that need to be addressed before considering your manuscript for publication, which motivates my decision of “Major revision”.

In preparing your revised manuscript, I strongly recommend you follow the suggestions and edits indicated by the reviewers. In particular, they have identified lacking methodological details, low readability of figures, limited contextualization and discussion of the biological/clinical/experimental implications of this work, lacking statistical or quantitative metrics to support some results, and limited comparison of the proposed approach to other alternative methods and software packages.

We will be looking forward to receiving your revised manuscript.

Best regards,

GL

**Journal Requirements:**

At this stage, the following Authors/Authors require contributions: Matthias Bruhns, Jan Thomas Schleicher, Maximilian Wirth, Marcello Zago, Sepideh Babaei, and Manfred Claassen. Please ensure that the full contributions of each author are acknowledged in the "Add/Edit/Remove Authors" section of our submission form.

3) We notice that your supplementary Tables are included in the manuscript file. Please remove them and upload them with the file type 'Supporting Information'. Please ensure that each Supporting Information file has a legend listed in the manuscript after the references list.

**Reviewers' comments:**

Reviewer's Responses to Questions

**Comments to the Authors:**

Reviewer #1: This study provides a rigorous quantification of segmentation errors and their downstream effects, offering a much-needed benchmark for understanding error propagation in single-cell analyses. The use of affine transformations to simulate realistic errors is innovative, allowing systematic evaluation of segmentation robustness. The integration of k-Means and Leiden clustering alongside Gaussian Mixture Models broadens the findings interest and clear visualizations (e.g., Figures 2, 5, 6) make complex results interpretable. The study’s practical implications extend to real-world spatial proteomics pipelines, emphasizing the necessity of improved segmentation evaluation metrics and quality control methods.

However, the authors reference a number of software tools that have previously been used to address the same or similar problems, including STARLING, CyLinter, and ESQmodel. This reviewer feel that the datasets generated by the authors could be run through those software to strengthen the findings in this manuscript and possibly to begin addressing some of the field issues the authors have noted.

Major Revisions

Probabilistic Clustering with STARLING. The authors could compare standard clustering approaches (Leiden, k-Means) with STARLING, a segmentation-aware probabilistic model. This could be done by running STARLING on the perturbed datasets and comparing the Adjusted Rand Index (ARI) and clustering stability against the current results (Figure 5). If STARLING improves clustering accuracy under segmentation errors, this would strengthen the study’s impact.

Quality Control with CyLinter. To assess the impact of quality control on segmentation artifacts, the authors could apply CyLinter to their perturbed datasets and remove flagged segmentations. By comparing clustering and phenotyping accuracy before and after artifact removal, they can quantify CyLinter’s benefit (Figure 6B). This would demonstrate whether quality control can mitigate the negative effects of segmentation errors.

Alternative Segmentation Evaluation with ESQmodel. The manuscript critiques IoU-F1 thresholding but does not provide an alternative. The authors could apply ESQmodel, which evaluates segmentation accuracy with biologically informed metrics. By correlating ESQmodel scores with phenotyping errors (Figure 6) and clustering robustness (Figure 5), they can determine if it provides a more meaningful segmentation quality metric than IoU-F1.

Minor Revisions

Scalability Considerations with MCMICRO. The authors should discuss how MCMICRO could enable large-scale segmentation benchmarking. Addressing its modularity and efficiency in processing high-dimensional imaging data would clarify the study’s applicability to larger datasets.

Probabilistic Correction for Phenotyping Errors. The authors should propose a probabilistic post-processing correction for phenotyping errors, similar to STARLING’s segmentation-aware approach. A Bayesian reassignment model could improve phenotype consistency, mitigating segmentation-induced misclassifications (Figure 6).

Reviewer #2: The authors demonstrate the effect of varying degrees of missegmentation on cell type calling in high plex tissues. The analysis is replete with comparisons of IOU and F1 scores and cell type clusterings. The intended audience appears to be computational biologists, image analysts, and data scientists. However, it might be challenging for an experimentalist to still fully appreciate the problem addressed in this paper. The data is present in the paper but it needs abit more discussion of the biological/clinical implications.

Major suggestions:

1. I suggest adding more figures with actual cells from dataset referenced in [27] with perturbed masks overlaid to help readers see precisely why/where/when masks are not overlapping with specific markers. This would add alot more weight to the paper.

2. I would not have initially thought of using translation, rotation, affine, etc as ways to perturb masks. Segmentation errors are not caused by shifting the masks around like that. A more intuitive transformation would be to do subtle warping, which many image analysis packages can do.

3. Line 360: “potentially misleading biological conclusions”: While this is true, it might be helpful to check with a clinician/biologist if this statement can be made more specific. We know that cell type calling can be inaccurate due to segmentation. How would that influence diagnosis of disease? Maybe the number of T cells infiltrating a tumor is lower measured leading to a wrong treatment outcome.

4. Line 368-370: This is a good idea but I’ve read many papers propose to model the segmentation error into their analysis but no one goes beyond to state more clearly how this would be done. What exactly would this look like? Please add 2-3 more sentences how an error model could be used and introduced.

Minor revisions:

1. Since this is a segmentation-centric paper, I suggest adding more references to the introduction on existing segmentation packages used for high plex imaging such as Stardist, UnMICST, MIMO-NET, and MICRO-NET since they all have different strengths and weaknesses.

2. Figure 1 shows example of an original vs perturbed mask for an entire tissue. These panels look identical to each other because they are zoomed out. A much more effective use of space is to show zoomed in versions where readers can actually see changes in mask shape. Also, scale bars should be added to all panels where applicable.

3. Please state more clearly whether all transformations (translation, rotation, etc) are done in combination or just one.

4. Line 160: you state that there are no comprehensive benchmarking dataset with ground truth. This is not true. UnMICST has manually annotated outlines of nuclei from human tissue.

5. Figure 2B: Suggest to overlay curves for all F1 scores for each marker in Figure 2B over each other for easier comparison.

6. Line 234: clarify if this is done on all cells in aggregate or specific cell type?

7. Fig. 3, 4, 5 : state how many cells/data points are represented by each bar?

8. Fig. 4 and 5: title ‘F1/preprocessing’ is confusing. What does that mean?

9. Fig. 6B & S4: What do the colors mean? Add a color map.

10. Line 317: ‘4% of CD T cells are incorrectly assigned’. Is this a typo? This is not seen in matrix.

11. Please discuss why immune cells appear to be the most affected, according to 6B. Immune cells are smaller than tumor cells. Could this be why?

12. You list CODEX and MIBI as examples of imaging modalities in introduction. These have wildly different imaging resolutions. Please discuss in what way significant segmentation errors might be for different resolutions.

13. Fig S04: panels need more vertical spacing.

14. Line 376: If interested, there are other perturbations one can do/discuss. Change resolution by downsizing, change bit depth from 16bit to 8bit, background subtraction.

15. Github landing page could use more information about running code.

Reviewer #3: Bruhns et al. describe the downstream effect of using inaccurate cell segmentation results in single-cell analysis. Cell segmentation of a tissue is done to add spatial depth to a single-cell data by linking phenotype to their protein expression. However, the author shows that errors from many segmentation methods can have downstream effects especially in clustering by cell type classification and neighborhood mapping of the single-cell data. The authors took precaution to create ‘realistic’ segmentation errors–called a perturbation mask–by a combination of 7 types of affine image transformation methods to mimic errors that may occur in segmentation using other more established methods. The authors also created and enhanced a parameter to evaluate the degree of this perturbation when visually compared to the ground truth data called IoU-F1. They investigate the accuracy and similarity of the neighborhood mapping and clustering using both Jaccard index and Adjusted Rand Index with different neighbors and F1 values. Finally, the authors used Gaussian Mixture Model-based phenotyping to measure how accurately the dataset segmented using different F1 values of perturbation classify the cell types of all cells in the liver dataset.

Overall, this is a timely and informative study as spatial transcriptomics as a field is rapidly expanding, and this study takes a systematic approach to probe how downstream analysis can be affected due to a variety of segmentation errors. While the text was generally easy to follow (with certain exceptions, noted below), the figures need substantial work for readability and significantly below publication standards. Often appropriate statistics are missing thus diluting and questioning the conclusions made by the authors. We have several comments and concerns, in no particular order, that should be addressed:

We could not find Code or a README file. For a study that is benchmarking, providing the code and related data is indispensable to the review process.

Unclear if the authors expect their work to be a “tool” for others to use or a conceptual point for others to keep in mind. We recommend the authors add this nuance to the introduction and discussion.

For all figures: please add full forms of all acronyms used either in the figure or figure caption. Similarly, all labels should be adequately defined. As it stands, the figures presented a significant cognitive burden while evaluating this manuscript. Also add scale bars to all figures used.

The argument for analyzing “phenotypes” (last two sections) is a bit weak as ground truth for cell-type labels are not possible with some markers. As such, this kind of analysis is no different from what is done in the rest of the paper, including the cluster assignment analysis.

The authors should consider supplementing figure S1 with real-data examples on how affine transformation changes a mask in addition to the squares.

For Figure 1, show pixel/cell to represent the pixel size as compared to the cell so that the one-pixel gap between cells can be created that is mentioned in both lines 54-56 and 147. We are curious how this would affect the error of the segmentation for cells that are very close to each other or overlap.

For Figure 1 again, there is a schematic of the pipeline created for this benchmark but showing a zoomed in picture of a group of cells (1-10) in brightfield and its masks would also help explain lines 149-154 to show how this ‘empirical selection process’ selects a ‘realistic error’. Furthermore, so we can visualize, F1 of the perturbation dataset in this figure can be stated.

Lines 118-125: Besides showing the equation and components of the equation, also explain how the ‘three components’ mentioned in line 307 fit into this equation. What does the N values represent from the gating strategy that is shown in Figure S3?

Spell out or explain HALO from line 165. We think it warrants an explanation as to why this one in particular is used as compared to other segmentation tools.

Figure 2A: Which one represents the original ground truth, an example with arrows or even a blow up version of one of the cells to show ground truth and perturbations, How did you compare the ‘visual appearance’ ‘to determine noticeable artifacts’? Point out what is the small versus significant difference between the red-lined mask and the white-lined mask, assuming the white is reference. In general, this figure needs significant work for the reviewer to understand the key points.

Figure 2D and Line 200. In the figure it shows as mean of absolute error of features while in the passage its stated as median. Which one are you referring to? Also, are there more explanations as to why Pearson correlation is used to calculate this correlation? This can be done in the Methods section.

Line 210: Please explain what the chance level is, is there a specific value for this threshold?

Line 233-234 shows a strong point to be made in the argument regarding the neighborhoods with distribution of IOU values which is proven with Figure S2. Placing it in Figure 3 would add importance to this information and be a stronger representation of the author’s explanation.

Line 253-255: What does it mean to “consider the most similar cluster assignment for ground truth and perturbed data”? Are all the ARI represented only show the ones with the highest ARI value. It is never stated what is the range of the ARI value (typically -1 to 1 where 1 shows highest similarity) but the images only show ARI above 0. Do the authors calculate ARI with ranges 0-1? Furthermore, please elaborate on how this shows that the differences observed are only due to “introduced perturbations”. Could the authors provide quantitative evidence for their claim? Possibly a way to distinguish the noise between a clustering error and perturbation.

Figure 4 is only referred once in the text without mentioning the subfigures (4A, 4B, 4C and 4D). Reading Lines 258-272 creates confusion as to which subfigures each conclusion is pointed to.

In each Figure 4 and Figure 5, subfigures A and C have the same caption. It is not shown or mentioned in the captions that there is a difference between the two is from application of log(x+1) transformation as a variance stabilization, nor which subfigure has variance stabilization or not. This ‘preprocessing variance’ step mentioned in Line 259 creates confusion as it is not stated again what the difference is in that paragraph nor shown in the graph.

Furthermore, Figure 5 A and C look identical, and Figure 5E does not have a label on its scale bar. How does the difference in color contribute to the claim in Line 291 that log(x+1) data affects clustering robustness? Please provide clear evidence to support this claim?

Overall when explaining differences between histograms of any metric no p-values are mentioned to show significance when comparisons are made. The comparisons by eye are of very little difference as also mentioned in Lines 292-294.

Lines 305-310: Please specify and elaborate on what are the ‘first two components’ and the ‘three components’ used to reduce the number of false-positive gates. It is mentioned that its in reference to the diagram in Figure S3, a smaller and simpler diagram can be made to Figure 6 from Figure S3 that clearly shows what the components are and what cell type they represent or will positively gate.

Lines 310-312: Please elaborate or use equations to represent what balance accuracy meant in this context and the range of values. Would 0.8 be a representation of -1 to 1 or 0 to1 or 0 to 2?

Figure S3: The caption only indicates what extra information is added to the figure but does not show where cell lineage or cell subtype that is mentioned in Lines 314 and 317-320 refer to. Please write an elaborate description on the gating and how the categories mentioned in the lines above are classified through the diagram.

Line 320: “we use Wu and Palmer distance (see)” there seems to be missing information in this sentence. The Wu and Palmer distance is not explained nor mentioned in the Methods section to show how it quantifies the different errors.

Line 286-287: Mentioned that with lower neighbors it has higher ARI values, which is not what is represented in Figure 5D. This also affects Line 289 as Figure 4D and Figure 5D have opposite trends, which is opposite of the claim that shows the result matches with k-means algorithm. Figure 5D the differences between each bar looks very small with no significance. Higher number of neighbors induces fewer clusters can be seen in Figure 5E but not from 5D.

Line 288: Please elaborate and show evidence on the claim that neighborhood size ‘amplifies sensitivity’ to perturbation-induced feature shifts.

Why have the authors not considered using one or two other segmentation tools on the same dataset and no perturbations on the metric as well?

A brief discussion on why CODEX was primarily chosen for analysis could be beneficial for the study and how their approach extends to other modalities and technologies.

In general, the authors are strongly recommended to expand their metrics for calculating errors and benchmarking for different modalities/aspects. In the single-cell RNA sequencing space, this has been thoroughly investigated for singlet/doublet detection by the following papers: PMID: 33338399 and PMID: 38925122. Perhaps a discussion on how the downstream interpretations are often affected and propagated in the field of scRNAseq (which was a precursor to modern spatial biology) through these mentioned papers and lessons learnt could be informative for inspiring the present study.

Reviewer #4: This paper employs a systematic perturbation framework to simulate realistic segmentation errors, using affine transformations and binary opening, and evaluate their effects on key tasks such as k-nearest neighbor (kNN) graph construction and unsupervised clustering (k-Means and Leiden). The study is timely and relevant. However, there are several areas need clarification:

1. The Results sections often present trends (e.g., ARI decrease with perturbation strength, Fig 4 and 5; balanced accuracy below 0.8, Fig 6A) without providing specific numerical values or statistical tests to confirm significance.

2. Several methodological details are missing or unclear, limiting the reproducibility. For example, the range of k values for k-Means and Leiden clustering is not specified.

3. The authors rely on visual inspection (Fig 2A) to determine that the perturbed masks maintain a consistent structure with the original masks, with "no apparent irregularities or distortions." However, visual inspection is inherently subjective and lacks quantitative rigor.

4. The authors interpret the classifier’s low accuracy as evidence of realistic perturbations due to high similarity between original and perturbed data, but, this interpretation may be confounded by alternative explanations, such as the classifier being underpowered or the selected features. I recommend discussing these alternative explanations for the classifier’s performance.

**Have the authors made all data and (if applicable) computational code underlying the findings in their manuscript fully available?**

Reviewer #1: Yes

Reviewer #2: Yes

Reviewer #3: **No: **See comments.

Reviewer #4: Yes

PLOS authors have the option to publish the peer review history of their article (what does this mean?). If published, this will include your full peer review and any attached files.

Reviewer #1: **Yes: **Francesco Pasqualini

Reviewer #2: No

Reviewer #3: No

Reviewer #4: No

**Figure resubmission:**
---

## [Decision Letter · Decision Letter 1]

22 Jul 2025

Dear Prof. Dr. Claassen,

We are pleased to inform you that your manuscript 'Effects of segmentation errors on downstream-analysis in highly-multiplexed tissue imaging' has been provisionally accepted for publication in PLOS Computational Biology.

Best regards,

Guillermo Lorenzo

Academic Editor

PLOS Computational Biology

Pedro Mendes

Section Editor

PLOS Computational Biology

Dear authors,

I am glad to let you know that all reviewers have agreed to recommend your work for publication. I would like to congratulate you on the work presented in the manuscript and I look forwards to seeing future developments from your team.

Sincerely,

GL

Reviewer's Responses to Questions

**Comments to the Authors:**

Reviewer #1: I appreciate the effort of the authors in trying to address the comparisons with existing related software packages but I understand their discussion of enountered technical challenges and/or the conceptual differences. I would still suggest that they consider additing some of these insights to the Discussion, but I leave this up to the authors and editors to decide.

Reviewer #2: Thank you - I think all comments have been addressed by revisions! Great job!

Reviewer #3: The authors have done an excellent job in their rebuttal. I also appreciate the respectful pushback on some of my suggestions -- their justifications are clear and agreeable.

Reviewer #4: The authors have addressed my questions and comments from the last review.

**Have the authors made all data and (if applicable) computational code underlying the findings in their manuscript fully available?**

Reviewer #1: Yes

Reviewer #2: Yes

Reviewer #3: None

Reviewer #4: None

PLOS authors have the option to publish the peer review history of their article (what does this mean?). If published, this will include your full peer review and any attached files.

Reviewer #1: **Yes: **Francesco Pasqualini

Reviewer #2: No

Reviewer #3: **Yes: **Yogesh Goyal

Reviewer #4: No

---

## [Editor Report · Acceptance letter]

PCOMPBIOL-D-25-00324R1

Effects of segmentation errors on downstream-analysis in highly-multiplexed tissue imaging

Dear Dr Claassen,

I am pleased to inform you that your manuscript has been formally accepted for publication in PLOS Computational Biology. Your manuscript is now with our production department and you will be notified of the publication date in due course.

With kind regards,

Judit Kozma
